# Exploring access to HIV-related services and programmatic gaps for Men having Sex with Men (MSM) in rural India- a qualitative study

**Sampada Bangar[1], Uday Mohan[2], Sanjeev Kumar[3], Amarendra Mahapatra[4], Shivendra Kumar Singh[2], Rewa Kohli[5], Archana Verma[5], Tuman Lal Katendra[5], Girish Rahane[5], Suhas P. Shewale[5], Nayana Yenbhar[5], Vinita Verma[6], P. Saravanamurthy[7], Bitra George[7], Bhawani Singh Kushwaha[6], Chinmoyee Das[6], Shobini Rajan[6], Seema Sahay[5]***

1 Epidemiology Division, ICMR-National AIDS Research Institute, Pune, India, 2 Upgraded Department of Community Medicine and Public Health, King George's Medical University, Lucknow, India, 3 Department of Community & Family Medicine, All India Institute of Medical Sciences, Bhopal, India, 4 Epidemiology Division, ICMR-Regional Medical Research Center, Bhubaneswar, India, 5 Division of Social and Behavioral Research, ICMR-National AIDS Research Institute, Pune, India, 6 National AIDS Control Organization, New Delhi, India, 7 FHI 360, Linkages-India, New Delhi, India

* ssahay@nariindia.org, seemasahay@yahoo.com

**Data Availability Statement:** The study was conducted among sensitive Hidden HIV Key population (MSM) hence the data sharing is restricted. Data will be available at the centralized

## Abstract

### Background

Despite the Link Worker Scheme to address the HIV risk and vulnerabilities in rural areas, reaching out to unreached men having sex with men (MSM) remains a challenge in rural India. This study explored issues around health care access and programmatic gaps among MSM in rural settings of India.

### Methods

We conducted eight Focused Group Discussions (FGDs), 20 Key Informant Interviews (KIIs), and 20 In-Depth Interviews (IDIs) in four rural sites in Maharashtra, Odisha, Madhya Pradesh, and Uttar Pradesh between November 2018 and September 2019. The data in the local language were audio-recorded, transcribed, and translated. Data were analyzed in NVivo version 11.0 software using the grounded theory approach.

### Results

Primary barriers to health care access were lack of knowledge, myths and misconceptions, not having faith in the quality of services, program invisibility in a rural setting, and anticipated stigma at government health facilities. Government-targeted intervention services did not seem to be optimally advertised in rural areas as MSM showed a lack of information about it. Those who knew reported not accessing the available government facilities due to lack of ambient services, fear of the stigma transforming into fear of breach of confidentiality. One MSM from Odisha expressed, "...*they get fear to go to the hospital because they know that hospital will not maintain confidentiality because they are local people. If society will know about them, then family life will be disturbed*" [OR-R-KI-04]. Participants expressed

data repository at ICMR-NARI and will be available from the Director, ICMR-NARI (director@nariindia.org) for researchers who meet the criteria for access to confidential data.

**Funding:** This work was supported by the United States Agency for International Development (USAID) through FHI 360/Linkages [Grant Number: AID-OAA-A-14-00045]. Corresponding author received the funds. The funders had no role in study design, data collection, and analysis, decision to publish, or preparation of the manuscript.

**Competing interests:** The authors have declared that no competing interests exist.

the desire for services similar to those provided by the Accredited Social Health Activists (ASHA), frontline health workers for MSM.

## Conclusion

Programme invisibility emerges as the most critical issue for rural and young MSM. Adolescent and *panthis* emerged as Hidden MSM and they need focused attention from the programme. The need for village-level workers such as ASHA specifically for the MSM population emerged. MSM-friendly health clinics would help to improve healthcare access in rural MSMs under Sexual and Reproductive Health Care.

## Introduction

Men who have sex with men (MSM) or gay men are disproportionately impacted by the HIV epidemic. The MSM population is 26 times more likely to be infected by HIV than the general population [1, 2]. In the year 2019, 23% of new HIV infections globally were among gay men / MSM [2]. The geographical importance of India is critical because the HIV prevalence among MSM might translate into large numbers. As reported by Elangovan et al, as of 2018–2019, there were nearly 5.7 lakhs estimated MSM in India [3]. The HIV epidemic in India is 'concentrated' among 'key populations' (KPs) who are engaged in unprotected sex with multiple partners or injecting drug users [4]. People with Injecting Drug use are concentrated in a small geographic area of India as compared to MSM who are spread all across the country. Hence MSM population becomes one of the key drivers of the HIV epidemic in India and has been prioritized as a core high-risk group for HIV prevention [5, 6]. Trends show stabilization of the HIV epidemic among MSM in India despite the targeted intervention (TI) programs being implemented specifically for this population. The HIV prevalence, among MSM, was 4.3% between 2011 and 2016 indicating stabilization of the HIV epidemic [6].

Under the National AIDS Control Program (NACP), HIV prevention services are provided through Targeted Interventions (TIs) and the Link Workers Scheme (LWS) in rural India where health services are active in 219 districts. Three implementation models viz. Focused, Transit, and Convergence models at the state and districts level have been implemented to address diversity in the epidemic in rural settings. TI program in India provides a basket of services focusing on HIV prevention, and support, and provides linkage to a high-risk vulnerable key and bridge population in defined geographies [7]. Non-Governmental Organizations (NGOs) and Community-Based Organizations (CBOs) use a peer-led, outreach-based service delivery approach for this. LWS covers high-risk and vulnerable key populations in the rural areas. It primarily focuses on demand generation for various HIV/AIDS-related services, linking target populations to existing services, and providing a stigma-free environment for sustained access to information and services to key populations (KPs) under the NACP of India. The LWS covered around 18.88 lakh people including 9,850 MSM population in 2020–21 [8]. However, despite this, gaps continue to exist for coverage of key and bridge populations under TIs and LWS [9]. As a part of TI, MSM is primarily reached through hotspot-based approaches and LWS primarily does not create any service delivery points but rather focuses on linkages to existing health facilities under various programs. Fear of stigma and discrimination makes it difficult for rural MSM to reach these facilities. Achieving the first 95 of 95-95-95 targets of UNAIDS necessitates reaching this Hard to Reach MSM [HR-MSM] population for HIV testing and linking them with HIV care as per their HIV status [10].

In India, the spatial invisibility of high-risk MSM might be sustaining the HIV epidemic as they prefer to remain hidden due to prevailing stigmatizing socio-cultural and religious norms targeting same-sex relationships [11–13] and the socio-economic status of some geographical settings. High population densities in urban areas lead to individual anonymity giving safe spaces for the MSM to network. On the other hand, rural areas are smaller settings, less densely populated, close-knit communities, which live in self-inflicted social isolation as they have to hide their identity and very limited scope of covert networking [14–17].

The link between urban and rural MSM networks was evident in a study from Gujrat where 10% of the 200 MSM reported having their first homosexual encounter with a co-villager [18]. Another study in the four talukas of a Bangalore rural district also reported 1,060 estimated MSM of the total 8 lakh population [19]. These rural MSM have been traced despite their lack of self-identity, hidden nature, and a "language" for expressing the idea of male-to-male sex [20]. There is evidence for the existence of a still considerable proportion of MSM who might be either hidden or invisible and need to be reached to facilitate health care access to this vulnerable population. The TIs and LWS are specially provided for key vulnerable populations such as MSM, Female Sex Workers (FSWs), and People Who Inject Drugs (PWID). However, the reported decade-long stabilization of the HIV epidemic among MSM in India indicates the existence of hidden populations that might not be accessing TI services [21, 22].

Under the LWS of the national program, MSM are focused mainly on spot-based 'TI' settings and health facilities [23]. Data on MSM networks and their structures in settings outside TIs are limited in urban and rural areas. Although remaining hidden is one's right but access to health care falls under the purview of a health program. One of the first global target goals is that 95% of People Living with HIV (PLHIV) should know their status by the year 2030 [10]. Although LWS is quite active in rural settings in India, to achieve this target, MSM, who remain hidden anywhere more so in rural settings in especially the socio-economically disadvantaged states of India, need to be given access to public health facilities, especially for HIV testing which is a major gap. We present a multi-stakeholders analysis of the facilitators and barriers to health care access among Hard to reach-MSM and try to understand the programmatic gaps in reaching them in the rural settings in some of the socio-economically disadvantaged states of India.

## Materials and methods

A qualitative exploratory study was conducted in four urban and seven rural districts of the four states of India. Maharashtra (MH) in Western India, Uttar Pradesh (UP) in Northern India, Madhya Pradesh (MP) in Central India, and Odisha (OR) in Eastern India were included. We are presenting the findings from seven rural districts of these states. Only one relevant quote from urban data is included to support the findings of rural sites.

In each region, the states were selected based on the prevalence of HIV/AIDS among the MSM population reported in HIV Sentinel Surveillance (HSS) [24] and consultation with the national program. HSS is India's first intervention that plays a key role in monitoring the level and trend of the HIV epidemic in different categories of the high-risk population at different locations in the country using bio-behavioral data collection. The activity is conducted on a biennial basis. We used data from this survey to select the study sites. Maharashtra and Madhya Pradesh were selected considering the high HIV prevalence among the MSM population, with 4.69% and 4.40% respectively. Uttar Pradesh and Odisha were selected as there were scarce data on HIV and associated behaviors among MSM. According to recent data, the HIV prevalence was 1.14% in UP, whereas it was 0.80% in Odisha. Karad and Ichalkaranji (MH), Angul, Sundergarh & Banki (OR), Barabanki (UP), and Hoshangabad (MP) were the seven

rural sites. The operational definition of the MSM was men who had male-to-male sexual encounters (oral or anal) in the last three months and belonged to rural study sites. All study participants were adults and of legal age. Overall 20 Key-informant Interviews (KIIs), 20 In-depth Interviews (IDIs), and eight Focus Group Discussions (FGDs) were conducted across four study areas in rural settings. All participants were recruited purposively and conveniently for the interviews and FGDs. Five in-depth interviews were conducted in each study area with adult HR-MSM (not reached by TIs ever). MSM participants were recruited with the support of trained community liaison officers who belonged to the MSM community and who were employed to recruit study respondents and key population leaders using the snowballing technique. A Community Advisory Board (CAB) comprising both direct and indirect stakeholders was also established at all sites that provided inputs on the topic guides, study tools, and participant recruitment plan. The following strategies were used for the recruitment of participants: 1) *Stakeholder engagement*: Indigenous informants are the final and most valuable source of information to the researcher attempting to identify hard-to-reach populations. We identified primary and secondary stakeholders for the study to facilitate recruitment. 2) *CBO Leadership engagement*: CBO leaders were a valuable source of information and support. This engagement enhanced the community sensitization, mobilization, and recruitment processes. 3) *Community Liaison Officers (CLOs)*: Employing personnel from the study community helped gain the community's confidence, build trust, and establish rapport. 4) *Cultural engagement*: Participating in the programs conducted by the LGBTQ community facilitated the visibility and acceptability of the research team. Five key informant interviews (KIIs) per study area were conducted with private doctors, NGO/CBO representatives, TI representatives (counselors, clinicians), ART counselors, and key community representatives respectively. Two focus group discussions (FGDs) per study area were conducted with *kothis* (sexually receptive MSM) along with *double-decker* (sexually receptive and penetrative MSM)) and bisexual individuals.

## Study tools & data collection

The topic guides were developed separately for KII, IDI, and FGD in the English language. The guides were then translated into local vernacular languages [Hindi (MP & UP), Marathi (MH), and Odiya (OR)] with the help of the study sites staff and confirmed for uniformity and correctness by the site Principal Investigators (PIs) of these study sites. The translated guides were pilot tested at respective sites to ensure comprehension and the need to rephrase the questions. Suggestions for refinement and techniques for conducting the interviews were shared with the study team at all the sites to arrive at a consensus. Topic items were rephrased in English first to bring clarity and then translated accordingly. The topic guides focused on characteristics of the MSM community, experiences with health care, barriers, and facilitators, and expectations of the community for health care service access and utilization. Three step process was held for capacity building of the research team which included the reading of questions and discussions with the team, theoretical training on qualitative research, and two mock in-depth interviews with the MSM. This helped in rehearsing as well as understanding the flow of the questions and also in evolving the guides. Face-to-face interviews or discussions were conducted for KIIs, IDIs, and FGDs and audio recorded by trained master-level social science researchers. Each interview or discussion lasted for 45–90 minutes.

## Data analysis

The audio data were transcribed verbatim, translated into English, and typed into Microsoft Word immediately. The translated electronic data from the sites were reviewed by two social scientists and the principal investigator for correctness, completeness, clarity, and

methodological issues, and provided feedback within five days on the requirement for repeat interviews. Repeat interviews were conducted by the sites in case of missing or the requirement of additional information. Data were also reviewed to ensure the emergent themes to explore further relevant questions/ prompts. These were included in subsequent data collection opportunities. The final translated data was entered into N-Vivo Pro 11 software for analysis. Before embarking on the analysis, a data analysis workshop on Qualitative data analysis was organized that was attended by the PIs and the representatives from all four participating sites to identify initial codes using representative segments. The iterative readings of the data and inductive analysis of the text lead to several themes that were studied and analyzed to form categories. Using the grounded theory approach [25], the final data analysis was completed.

## Ethical considerations

The study protocol, participant information sheets, and informed consent form and process were approved by the Institutional Ethics Committee of the Indian Council of Medical Research (ICMR) -National AIDS Research Institute (NARI-EC/2014-13), Technical Resource Group (TRG) for Research (NACO), NACO Ethics Committee, Protection of Human Subjects Ethics Committee (PHSC) of FHI 360, and also the Ethics Committees of all the participating research institutes. During rapport-building discussions, Masters Level trained counselors recorded participants' legal age, ensured their comprehension verbally, and also checked that they were not under the influence of alcohol or any other recreational drugs. The counselors evaluated for coercion of any form and their alert state of mind to answer any question voluntarily. The consent was given to participants for reading or was read and explained to the illiterate participants. Counselors explained to all the participants about the study objectives, procedures, risks involved in participation, and measures the study team will take to minimize the risks involved in this study. They were informed about their right to withdraw or refuse to answer any question. Participants were given the freedom to consult peers, family members, or their physicians regarding participation in the study. They were also informed that their decision to not participate will not affect the research teams' relationship with them. The participants were always encouraged to ask questions and time was allotted to the participants after the interview for further questions, clarification, or for sharing any questions or doubts. On giving the opportunity to ask questions, they asked questions for clarification. During the informed consent process counselors confirmed with each participant about his comprehension of the study and his willingness to participate. Once the participant confirmed, written informed consent for participation and audio recording was obtained. The informed consent process and the interview were conducted in the local vernacular language to ensure clarity for the participants. The engagement of a community liaison officer who belonged to the local MSM community ensured the protection of the rights of the community and prevented the possibility of any coercion.

**Confidentiality and privacy.** The study was conducted in a sensitive population hence to maintain the confidentiality of the study participants, they were given unique Participant Identification (PID) numbers. Each participant underwent an informed consent process. Data were collected only after obtaining written informed consent. After the data collection, all data were kept under lock and key at each site. Data were transferred on the desktop which was password protected. Only authorized and limited site staff had an access to the data which were anonymized.

## Results

Twenty IDIs, 20 KIIs, and 08 FGDs which included a total of 42 participants were conducted in rural districts of the four selected states of India. The mean age of the MSM who

participated in IDIs and FGDs was 24.5 and 32.7 years respectively. Seventy percent of the FGD participants reported staying in a joint family, and 90% of IDI and 76% of FGD participants were unmarried. The emerging themes were 1) Wanting to remain hidden/ invisible, 2) No risk of acquiring HIV, 3) Not perceiving the need for HIV testing, 4) Lack of awareness about programs in rural areas, and 5) Program gaps: dissatisfaction with the existing services. For effective utilization of the services, most of the MSM felt the need for an MSM-exclusive (confidential) health facility led by 'community literate' healthcare providers in the program setting.

The emerging theme of 'Wanting to remain Hidden/ invisible' is explained as follows:

## Wanting to remain hidden/invisible

Rural districts form a small close-knit society with limited privacy and MSM felt tabooed from society. To conform to the social norm, most of the participants were reportedly married and working, which put pressure on them to remain invisible:

*"I know this type of MSM, who does not wish to come out in the open. They are very much interested in doing sex with MSM and they think that they are married people. . .Many MSM are there, who do the job in reputed organizations and think about family".*

[OR-R-ID-06, Odisha]

The taboo is ingrained and obviously, families do not reconcile as an MSM from Central India shared the reaction of the family when he 'came out':

*"Recently I have 'came out' to my family, so I have told my family about 'this' [/MSM identity/]. My family said that they will take me to the doctor [as] you are suffering from some disease."*

[MP-R-FGD-O3-05, Madhya Pradesh]

For those who are visible, the stigmatization is severe and they shared the stress of trying to protect their family from embarrassment.

*"When five-six people are sitting in our village, if we sit with them, then they whisper in our backside and told "Maichia". Why did God make us like this? Our life is sanguinary. We compromise with family, we suppress our feeling, so that the family accepts us."*

[OR-R-FGD-01, Odisha]

The non-stereotype MSM, for example, a *panthi*, is not visible in the community.

*"People call me for parties. Especially in villages. That too only kothis. No one brings panthis since others get attracted to them too. That's my rule. I never take my panthi along with me anywhere. Among ourselves too there is a lot of jealousy. There are those who even say to leave the other person and go to them."*

[MH-R-ID-01, Maharashtra]

Strategies to reach hidden MSM were discussed where the role of the key person and gaining access to their networks emerged as critical:

*"Directly you cannot approach them because they will not admit that they are MSM. We are also MSM, but through "Aisha Didi" [/key person/] you reached near us. You cannot directly reach hidden MSMs because they live normally in society. They are hiding their identity from society. You have to find their close ones and if you have found their network system, then maybe you can reach them. We know Aisha didi, and for her, we came near to you."*

[OR-FGD-01_R01, Odisha].

A front cover of socially acceptable jobs is used to remain hidden as MSM. A hidden MSM narrated as follows:

*"For the time being, they will be doing 'tie & dye, bandhej'*

[/a method of dying of stoles/].

*There are so many people who are coloring stoles. They are involved in sex work. They do this secretly".*

[UP-R-IDI-01, Uttar Pradesh]

To remain hidden, it was obvious that HIV testing would be a challenge. Not only being hidden but also the denial of self-risk and misconceptions seem to affect access and utilization of HIV testing. Therefore the next emerging theme was 'No risk of acquiring HIV':

## No risk of acquiring HIV

Denial is a challenge that needs to be addressed as it puts MSM at higher risk due to a false sense of security. Firm belief and faith that Kothis are invincible was a common observation across all sites. Another misdirected belief was about sexual partners from higher socioeconomic strata and younger age groups.

*"Kothis never get AIDS. It happens to only those who go to the 'narans' the girls."*

[MH-R-ID-01, Maharashtra]

*"I know that I don't have this type of disease [/HIV/] because I always do sex with high socioeconomic people. . . I never did sex with an aged person who might have the disease. Whenever I want sex, then I do with 15–20 years old people."*

[OR-R-ID-06, Odisha]

Not having faith in the health care providers and denial of HIV risk in different forms was a recurring phenomenon in all four rural sites.

*"But when we come out of the clinic, it comes in our thoughts that maybe the doctor is lying that we have HIV, or else the report is wrong*

[MP-R-FGD-04, Madhya Pradesh]

Even though HRMSM from rural India said that they had heard of ART or HIV but no one talked of having received information from reliable health sources such as healthcare providers or TI. They quoted newspaper advertisements, or the media as their source, and above that everyone believed that they cannot get HIV infection indicating low self-risk perception:

*"Our elder told us that the kothis are never infected even if we do it [/have sexual relations/] seven-eight times a day and whatever is there, it gets lost in the semen only."*

[MH-R-ID-01, Maharashtra]

The low-risk perception gained strength when MSM talked about multiple partners. The attitude seemed to be pleasurable and fun for them.

*"Yes, it changes. It's like today I have sex with someone two-three times; then, let's go with someone else for tomorrow. It works that way"*

[MH-R-ID-03, Maharashtra].

All the risk behaviors and ignorance lead to the new theme of 'Not perceiving the need for HIV Testing':

## Not perceiving the need for HIV testing

**HIV testing barrier-young MSM.** The risk of HIV acquisition persists in this population. Temporally the risk period ranges from several years. The concern emerges as the MSM reported that same-sex activity is initiated at a very young age. The act of sex initiation among the younger age group is mostly forced/ coercive which is repeated many times. These repetitions seemed to transition the boys into same-sex behavior. At the time of initiation, the boys were unaware of any risk involved in these acts.

*"I am 24 years old, I am a sex worker, and I am bottom, which means MSM bottom. I like being bottom in sex work, since [/I was/] seven or eight years old, then I had my first sexual contact when I was in second std. [/second grade at school/]. I had oral sex with my partner at that time. From then I had a habit. I was always been such girlish, so he [/partner/] realized that I am girlish, so he had sex with me, and even I was ready"*

[MH-R-KI-05, Maharashtra]

Another respondent reported four years as the age of sex initiation (possibly forced). The younger age is a barrier as they might not have the capacity to protect themselves or understand. Further, the fear of HIV test and doubts prevented MSM from getting an HIV test done. Not accessing HIV services was very common and most of the participants shared their abject fear, low motivation, and denial as the reasons for not going for HIV testing. In an FGD, the fear was simulated to a ghost:

*"Actually the thing is that we think that if there is that school in front [/pointing towards a building/], [/we/] do not go there at midnight as there are ghosts [/if a school is rumored to be haunted, so obviously we won't go there at midnight/]. A similar thing is there when we go for the [/HIV/] testing. . . That is why we do not go for the [/HIV/] testing, we are afraid of it"*

[MP-R-FGD-04-01, Madhya Pradesh].

**Lack of awareness about HIV testing services.** The lack of awareness about existing voluntary HIV testing and counseling centres among the study participants emerged. A key informant from Madhya Pradesh brought the invisibility of the program eloquently:

*"About HIV, I have only heard that it happens. Now I do not know how it [/HIV/] happens [/gets transmitted/] and why it happens. Because I haven't done anything like that [/risky/] for which I have to go to see a doctor. If it was [/risky/], then I would have told you. No madam I haven't heard. But, yes, I have read [/about/] it in the newspaper, otherwise, I haven't heard of it. Here the venereal disease doctor comes from = Bhopal = [/Capital city, MP/], I have seen his advertisement in a newspaper. Otherwise, I had not heard of it"*

[MP-R-ID-07, Madhya Pradesh]

Some of the participants were not aware of the existence of an HIV testing facility.

*"There is no such center [/HIV testing center/] here, and if so then I am not aware of it. This test [/HIV test/] may happen in the government hospital but I do not have much knowledge about it"*

[MP-R-ID-10, Madhya Pradesh]

*"They don't know that so many NGOs and CBOs would help them. So they take local home remedies. They don't take any treatment from the hospital. . .If a small health problem happens, it is ok but if some big health problem happens then they hide that finally, it turns HIV/ AIDS."*

[OR-R-KI-04, Odisha]

Since the TI sites are known to serve the key populations that are stigmatized; the same target community avoids accessing health care services from TI for fear of being recognized. To avoid this barrier, respondents suggested exclusive spaces for HIV testing and allied services for the MSM population.

*"- ma'am what happens is, that it is mainly associated with the gay community. Like there [/TI site/] these people also come and if we go there and four or more people are standing there, then they will recognize us. . .To increase acceptance and utilization of program services, Ma'am it should be like that only they [/TI/] keep things private, they should not talk about things like what is wrong with you, why have you got tested? I mean there should be no pressure on him like if HIV has happened then why it has happened and why have you gone there at that time? So . . . this should not happen there then we will get ready for doing the [HIV] test."*

[MP-R-FGD-04-03, Madhya Pradesh]

*". . .and if it [/TI/] is in a separate place then it will be better. . .I mean it should be a government place but it should be a bit separated [/located to manage confidentiality/]"*

[MP-R-FGD-04-02, Madhya Pradesh].

**Not so 'Safe sex' practice gives rise to a false sense of security.** Safe sex or protection is the common parlance in the field of HIV and other sexually transmitted diseases. The common assumption would be that these phrases are well understood by the key populations. However, the correct meaning/ practice of safe sex is not understood by the invisible rural MSM population at all in all states. As discussed earlier, they are very young and hidden to

prevent any identification. A false sense of protection due to the belief in following 'Safe sex' practices was cited by the rural MSM.

> *"No. Till now I don't get any type of problem and always I use jelly for safe sex and if I will get any kind of injury then I will manage"*

[OR-R-ID-06, Odisha].

> *"Yes they use saliva and, what say, like soap or oil, any smooth thing if didn't get a condom, otherwise use a condom".*

[UP-R-IDI-01, Uttar Pradesh]

> *"Yes, I knew him. He suggested using loks [/a local cream/]. . .before that, I used to use parachute oil [/brand of coconut oil/]."*

[MH-R-ID-01, Maharashtra]

MSM from the eastern state of rural Odisha had already shared a lower risk if partners were from higher socio-economic strata or younger age groups.

## Lack of awareness about programs

Ignorance about the existing program was observed. Under the National Program, HIV testing and counseling facilities are present in different geographical settings including rural areas. However, a key informant from Odisha pointed out the low visibility of the existing facilities. He shared his concern that if the information is not reaching the key population in urban areas, it has even higher implications for rural areas.

> *"There are many slum areas in = Bhubaneswar = [/capital of Odisha/] that they do not have access to the information. Although the government has informed free services, still they do not know where they could exactly avail the free condom or free HIV tests. NACO is doing the awareness program but they are doing it in a few places, therefore, the information is not penetrating to other places. . .so they do not repeat the same program everywhere. Therefore the information is missing. So until the information is passed through social media like TV or Radio, all this information cannot reach all places. This is the one [/area/] where the government is lagging behind".*

[OR-U-KII-03, Odisha]

As discussed earlier, despite the presence of LWS, it was noted that HIV services had no visibility in rural areas, reflecting the poor visibility of the program that needs to be addressed.

## Program gaps: Dissatisfaction with the quality of services

The first critical gap in the program emerged as no focused village-level worker for the MSM community. The health workers focus on several populations but no service is earmarked for MSM. A key informant stated:

> *"In rural areas, the situation is very worst because they [/MSM/] are uneducated. They [/MSM/] don't know anything. Government always provides help to mothers, girls, and transgender but the government never provides any health services to MSM"*

[OR-R-KI-04, Odisha]

Participants who had ever visited a health facility expressed their dissatisfaction with staff behavior, treatment received, and waiting time. Most of these were either from self-experiences or peer experiences. Fear of being dismissed, the non-empathetic behavior of staff at health facilities, and the non-availability of knowledgeable counselors were the factors that limited access to care. Almost all the participants from these rural sites acknowledged their preference for private health facilities over government health facilities because of their own or peer experiences concerning the quality of services at government hospitals.

*"Well, the service in a private hospital is good. If I go to a government hospital it will get unnecessarily advertised. We don't get good treatment too over there in the government hospital."*

[MH-R-ID-01, Maharashtra]

Inadequate information about HIV testing services influences the beliefs of the MSM community about government HIV services. Lack of trust in HCPs, fear of disclosure of identity, fear of breach of confidentiality, and quality of services received by government health care facilities were the emerging factors for not accessing the government HIV services.

*"Because they [/MSM/] have the fear that if they get their registration done, then, their name will get disclosed. So, because of this, they are afraid. For privacy reasons they do not avail the services there."*

[MP-R-KI-05, Madhya Pradesh]

Societal stigma, and saving the family from any embarrassment and shame results in the 'hidden' nature of MSM who are very young and need care. Most participants reported the perceived threat of disclosure and breach of confidentiality in HIV testing services centers.

*"For society and as we are staying in a village, we cannot go for HIV testing because all are known faces here: hospital staff, also local persons. They will ask why I am doing the HIV test"*

[OR-R-FGD-04, Odisha]

Therefore, the discussants also shared their expectations for the special clinic in special locations:

*"If the government will not provide us with special health facilities, it is ok. But we want a clinic in our area. Then also we can access our health facilities from that clinic. If someone will counsel us to understand our problem, then we will get much more benefit"*

[OR-R-FGD-01, Odisha]

TI sites are specifically meant for the MSM community rendering MSM self-conscious and they tend to avoid going to TI for any service.
Some spoke about a possible breach of confidentiality by the TI staff during village gossip:

*"We are afraid that they are asking for our home address, and father's name. Many people do not speak in front of us but they do it when they are with other people this boy is like this, he went for the test, went for it, he meets guys".*

[MP-R-FGD-04-03, Madhya Pradesh]

*"As we go to the hospital other people are there, we cannot show our private parts to the doctor because there are other people around, we need privacy. If the doctor will give us privacy then it will be more beneficial to us."*

[OR-FGD-01, Odisha]

## Needs of the community regarding place, and experienced personnel for an effective program

As the community explained their challenges, they also shared their views on program components that could be a better facilitator. The community expectations from the program were that they require a specific person or worker to cater to the specific needs of MSM.

*"When we get any injury in our anal region or sometimes bleeding occurs from the anal area, at that time we cannot tell to the doctor so we just wipe the blood with a cloth and apply coconut oil to that anal area. After that, if the bleeding is not controlled then we go near to the doctor and tell lies to the doctors because we cannot tell the truth. Like "ASHA Didi" [/Village level worker/] for females, if the government will provide healthcare staff who will understand us and understand our problem then it will be very helpful."*

[OR-R-FGD-1_R-4, Odisha]

The location of the health care facility was voiced repeatedly as a need. 'Urban location' for the facility for rural MSM emerged as a viable option to prevent the breach of confidentiality.

*"If you give health facility where my family members cannot reach that place and it would be far away from my home, then it will be better like a city area. And the doctor will check properly where we will be comfortable with them and they will listen properly about our problems".*

[OR-R-ID-06, Odisha]

## Counseling needs-understanding the approach of peers

A lack of skills among HCPs to understand the issues and needs of MSM were reported. Participants from rural areas voiced the need for a community-based health facility.

*"...there the main role is of counseling. The community member whether he is kothi or panthi or DD [/Double Decker/] whatever he is, so he [/MSM/] can share everything with the counselor freely. The counseling is the only session where he [/MSM/] can share all his views. For that we have to give them [/MSM/] a special time, we have to give the time according to their needs, we have to meet them according to their comfort."*

[MP-R-KI-09, Madhya Pradesh]

Community engagement by facilitating peer counseling was another expectation. MSM were aware of community representatives working in other public health programs for providing care like Accredited Social Health Activists (ASHA) or Multipurpose Workers (MPWs). Community-led health provision was found to be acceptable to MSM as reported by KI.

*"Definitely I will go to the doctor. I want that ASHA should be there for us, like others with whom we can share our problems regarding our disease. If somebody would be there from the*

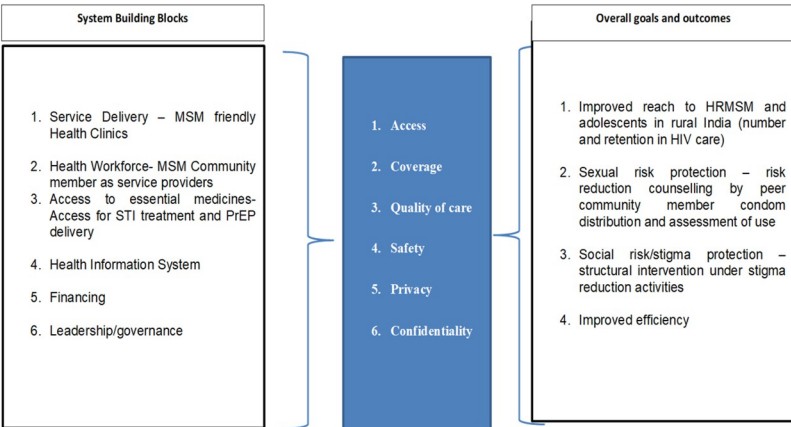

**Fig 1. Model adapted from WHO to improve MSM's access to health care in rural India.**

*MSM group, then we would be comfortable. If somebody else would be there, then, we would not be able to express our issues."*

[OR-R-IDI-02, Odisha]

*"Change what, we just want everything fine. Good counseling to be done, and good conversation to be done. All we just want is a little better attitude"*

[UP-R-IDI-01, Uttar Pradesh]

One of the study sites mentioned marriage as a facilitator for confidentiality which did not emerge in any other sites.

*"Marriage was one of the other reasons that made them not disclose that they were MSM. They felt 'getting married' was the facilitator of their confidentiality but experiences were varied".*

[OR-R-ID-06, Odisha]

To achieve a sustainable response to HIV, addressing the needs of the key population is critical [26]. Hence considering the WHO's framework of six building blocks of a health system [27] the following model (Fig 1) emerged to improve access to the health care system in rural India for MSM.

## Discussion

To the best of our knowledge, this qualitative study is one of the first studies among hidden and rural MSM in four states of India that explored the facilitators and barriers to their healthcare access, and the programmatic gaps. Strategic planning in the location of HIV testing services such as TIs, the deeply ingrained culture of confidentiality protection, and targeted information, education, and communications emerge as key pillars of addressing barriers to access and health care service utilization among MSM in rural settings. A very young age of introduction to sexual activity which must have been coercive in comparatively conservative societies of rural India is a matter of great concern. This calls for community-based structural interventions for the protection of young boys. Prevention programs need to be implemented

at young age possibly at the school level. Protection and education of 'boys' emerge as a need in a country where the male gender is considered strong, not requiring protection. In 2007, the Ministry of Women and Child Welfare, supported by the United Nations Children's Fund, Save the Children, and Prayas, found that 53.22% of children faced one or more forms of sexual abuse; among them, the number of boys abused was 52.94% [28]. It is critical to focus on young boys, especially in India where male rape and sexual abuse of a male child although legally a crime; but socially, is kept hidden due to the hegemony of patriarchy and ensuing attitudes towards masculinity [29]. Young adolescent boys, *panthis*, and MSM afraid of the stigma and the possibility of breach of confidentiality remain as hidden/ hard-to-reach MSM. Enabling an environment for greater acceptance of sexual orientations is needed to be inculcated at a younger stage of life. This might help in mainstreaming the otherwise stigmatized population. Homophobia against the 'visible MSM population' differentially influences the HIV/ STI care needs of MSM. Homophobia needs to be addressed through structural interventions to bring acceptance not only in the community but also in the HCPs by mainstreaming the population.

Geographical differences in the utilization of HIV testing and prevention services are evident. Issues of the invisible/ hidden key population in a rural setting especially among MSM are different than those in urban settings. Although, India sets a strong example of a TI program for key vulnerable populations; the visibility of interventions especially targeting MSM emerges as a gap in rural India. LWS focuses on identifying and reaching networks of high-risk vulnerable populations in rural settings. However, large geographical rural areas and limited resources limit the optimal reach of this scheme in rural settings. Respondents from rural settings in two of the states did share about MSM's unawareness of the program. In addition to low-risk perception, other barriers included sub-optimal quality of care at government facilities, stigma, discrimination, and fear of breach of confidentiality. The need for a safe private place for discussing community issues and genital examination by experienced health care providers belonging to the community, and counseling were voiced strongly by the community.

After the proven efficacy of Pre-exposure prophylaxis (PrEP) [30], FDA has recommended the use of PrEP in high-risk populations for the prevention of HIV [31, 32]. HIV testing provides an important entry point to such bio-medical HIV prevention strategies including PrEP and also targets the first goal of the End AIDS epidemic [10]. However, cited reports of past sexual abuse at a very young age (four and eight years) when they do not know anything about HIV is a matter of concern for the linkage to HIV care services among MSM. It is already known that an early sexual debut poses potential health risks. In a study conducted among young MSM, the younger age group reported more exchange sex; drug use; emotional/ psychological problems, history of suicide attempts, and contracting HIV with an increased risk of unprotected intercourse [33]. In our study, many MSM were either unaware or in denial about the risks of same-sex behaviors among men. Lack of awareness of services and protective interventions were other critical barriers faced by the hidden, young MSM in rural settings in India. Similar findings were reported from studies conducted in Myanmar [34], Thailand [35], and Peru [36].

Stigma is universal for MSM which limits them from accessing health care and similar findings were reported in South Africa [37, 38]. In our study too, stigma forms the key social determinant of poor or no access to HIV care services among rural MSM. In rural India, the fear of breach of confidentiality is likely to be more pronounced for which safer spaces away from denser areas, are expected from the program. The abolition of section 377 [39] has not changed the attitude of the community or the law enforcers and it continues to force an MSM to hide. Most participants who took part in our study expressed conditional willingness to get

registered with TI, only after the assurance that their identity will not get revealed. The assurance on the degree of anonymity drives the health-seeking behavior among hidden MSM and hence needs to be addressed sensitively. Development of structural interventions under stigma reduction activities, and clinic setups ensuring privacy and confidentiality are needed for acceptance of MSM to improve HIV testing and health care access. Strategic planning for improving the visibility of the services is needed for optimal utilization of health care services. Electronic and /or media sources mentioned by rural MSM can be a potential strategy to advertise government programs and services in villages.

Community mobilization is important to understand issues of MSM in the community. Village-level healthcare workers such as Accredited Social Health Activists (ASHA) designated specifically for MSM-related issues emerged as a need. NACP needs to focus on programs tailored for MSM in rural areas/ settings specifically focusing on hidden MSM who are left behind. Expansion of prevention and IEC services for HRGs which are more hidden in both urban and rural areas form one of the key strategies of NACP IV. However, while designing new strategies, a community-based person needs to be involved for empathetic understanding and redressal of their needs. Our study findings give new insights into the unidentified MSM network that might not have been covered under TI settings. One of the crucial emerging groups is very young adolescent boys. Further, this study generates information on the MSM network, mostly hidden, in rural settings for the first time in India which is indicative of a need for culturally and locally acceptable strategies for HR-MSM in the rural area.

The key informants raised their concerns about the paucity of targeted HIV prevention and treatment services for MSM in rural areas. In many places, there was no TI site although the key population did exist indicating the need for new mapping. The lack of less-equipped TI services was another concern shared by the participants. Non-recognition of *panthis* (MSM who practice penetrative sex or both penetrative and receptive sex) in the TI program got highlighted as these are the most at-risk hidden MSM in the community as indicated in previous studies too [40, 41]. In our present study, participants did bring their sexual partners but it was a challenge. Godbole et al. recommended focusing on a targeted intervention program for MSM who show penetrative behavior with other men, which could help in reaching this population [40]. The partners of kothi are the hidden MSM, not linked to care indicating the need for a unique targeted strategy to reach masculine sexual partners of the MSM. These men might be a new group of bridge population adding to the stabilization of the HIV epidemic being observed in the country.

The next emerging issue was the positioning of healthcare services. Although there were varied responses on whether health services for MSM should be mainstreamed within the public sector, or whether separate services were needed; overall, there is agreement on the need for some specialized services for MSM. The engagement of the community was voiced clearly. MSM conveyed their preference for community people providing services in such health centers for them. There is a need for strengthening government capacities and available systems for support, and this is a critical output of the analysis. The program needs to focus on building the capacities of the healthcare providers for MSM ambient facilities.

Participants in our study reported initiation of same-sex activities in their adolescent years when they are afraid and felt constrained. Therefore, an ambient environment for youth is required where they would come forth with their issues such as confusion about feeling attracted to the same gender; being forced, and having succumbed to pressure. A socio-culturally acceptable intervention is needed to target adolescent age groups. An adolescent might not acknowledge their orientation, avoid thinking about it or come up with an alternate explanation for their feelings [42] but improving risk perception is critical in improving acceptance of biomedical HIV prevention strategies such as PrEP [43]. Community mobilization and

structural interventions for the protection and empowerment of young adolescent boys are recommended. Adolescent-friendly health clinics for all adolescents can address some of these issues [44].

Integration of programs emerged as a strategy for optimal utilization of program services. For effective reach to rural MSM, in addition to improving program visibility, the service delivery needs to be shifted to "MSM-friendly health clinics" under Sexual and Reproductive Health Care. The primary focus of information would be "addressing health issues of same-sex behavior ensuring privacy and confidentiality". As recommended by most participants a knowledgeable staff belonging to the community is needed under the 'Health Workforce'. Considering this, an already existing ASHA model can be replicated for the MSM population which could be conducively instrumental in reaching the community. Uninterrupted access to medicines for the treatment of STIs at these clinics and PrEP delivery also can be considered through these clinics if the model shows efficacy. The involvement of outreach workers and peer educators from NGO/ CBO working with the community has already been used in the program as a useful strategy for reaching out to hidden MSM and linking them to HIV care. However, expectations were voiced for key population-based community outreach workers. The primary objective of restructuring existing interventions/ schemes should be on improving: 1) Access, 2) Coverage, 3) Quality of Care, 4) Safety, 5) Privacy, and 6) Confidentiality. The overall goals and outcomes would then include improved reach out to HR-MSM that should include *Panthis* and adolescents. We suggest feasibility testing of this model through implementation research for improved efficiency of the existing program in reaching HR-MSM.

## Strengths and limitations

This was a qualitative exploration of facilitators and barriers to healthcare access among four rural districts covering four regions in a second-populous country like India. In-depth interviews helped in understanding the grass root challenges of the community and feasible solutions to overcome them. Although the data presented in this paper represent four rural districts of India, the generalizability is limited. However, an in-depth identification of programmatic gaps in healthcare access among rural MSM paves the way forward for hidden and invisible populations and can be tested further.

## Conclusion

The MSM face pressures of remaining invisible in less dense settings of rural India. A close and watchful community makes them conform to social norms by getting married. The initiation of MSM behaviors at a very young stage spells sustained risk with no program to address the problems of adolescent MSM. Attaining adulthood brings not much relief as the study participants professed ignorance of the program. The program's invisibility, misconceptions and denial of HIV risk, suboptimal quality of services, and anticipated stigma at government health facilities emerged as barriers to healthcare access among MSM in rural settings. There is a need to include *panthis* both as a new bridge and MSM key population in the program along with designing strategies to reach them to be formulated and tested. Adolescent-friendly health clinics are recommended for providing sexual and reproductive health education and to help acknowledge and respect the existence of various sexual orientations.

## Acknowledgments

The study was led by ICMR-National AIDS Research Institute, Pune. We acknowledge the support received from Director, ICMR-NARI for the entire study. We are thankful for the

continued support received by the Indian Council of Medical Research, India. We sincerely acknowledge the support received from the National AIDS Control Organization, India right from the inception to the conclusion of Phase 1 of the study. It was critical to receive support from State AIDS Control Societies of the respective states which were facilitated by NACO. The entire study team expresses heartfelt thanks to our study participants without whom this study would not have been completed. We sincerely acknowledge the support received from Professor Prashant Kumar (Retd), Head of Department, Department of English, SK University, Dumka, Jharkhand, India for copy editing and grammar corrections.

## Author Contributions

**Conceptualization:** Seema Sahay.

**Data curation:** Sampada Bangar, Rewa Kohli, Archana Verma, Nayana Yenbhar, Seema Sahay.

**Formal analysis:** Sampada Bangar, Sanjeev Kumar, Amarendra Mahapatra, Shivendra Kumar Singh, Rewa Kohli, Archana Verma, Suhas P. Shewale, Seema Sahay.

**Funding acquisition:** Seema Sahay.

**Investigation:** Uday Mohan, Sanjeev Kumar, Amarendra Mahapatra, Shivendra Kumar Singh, Rewa Kohli, Archana Verma, Tuman Lal Katendra, Girish Rahane.

**Methodology:** Seema Sahay.

**Project administration:** Sampada Bangar, Uday Mohan, Sanjeev Kumar, Amarendra Mahapatra, Shivendra Kumar Singh, Seema Sahay.

**Supervision:** Sampada Bangar, Uday Mohan, Sanjeev Kumar, Amarendra Mahapatra, Shivendra Kumar Singh, Rewa Kohli, Archana Verma, Seema Sahay.

**Validation:** Sampada Bangar, Rewa Kohli, Archana Verma, Seema Sahay.

**Visualization:** Sampada Bangar, Rewa Kohli, Archana Verma, Seema Sahay.

**Writing – original draft:** Sampada Bangar, Sanjeev Kumar, Amarendra Mahapatra, Shivendra Kumar Singh, Rewa Kohli, Archana Verma, Seema Sahay.

**Writing – review & editing:** Sampada Bangar, Sanjeev Kumar, Amarendra Mahapatra, Shivendra Kumar Singh, Rewa Kohli, Archana Verma, Tuman Lal Katendra, Girish Rahane, Suhas P. Shewale, Nayana Yenbhar, Vinita Verma, P. Saravanamurthy, Bitra George, Bhawani Singh Kushwaha, Chinmoyee Das, Shobini Rajan, Seema Sahay.

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
