## [Decision Letter · Decision Letter 0]

23 Aug 2022

PONE-D-22-17552Exploring health care access and programmatic gaps for Men having Sex with Men (MSM) in rural India- a qualitative studyPLOS ONE

Dear Dr. Sahay,

Thank you for submitting your manuscript to PLOS ONE. After careful consideration, we feel that it has merit but does not fully meet PLOS ONE’s publication criteria as it currently stands. Therefore, we invite you to submit a revised version of the manuscript that addresses the points raised during the review process.

We look forward to receiving your revised manuscript.

Kind regards,

Jalandhar Pradhan

Academic Editor

PLOS ONE

Journal Requirements:

2. You indicated that you had ethical approval for your study. Please clarify whether minors were involved in your study. If so, in your Methods section, please ensure you have also stated whether you obtained consent from parents or guardians of the minors included in the study or whether the research ethics committee or IRB specifically waived the need for their consent.

3. Please describe in your methods section how capacity to provide consent was determined for the participants in this study. Please also state whether your ethics committee or IRB approved this consent procedure. If you did not assess capacity to consent please briefly outline why this was not necessary in this case.”

4. Please provide the full names of all ethics committees that approved this study.

"This work was supported by the United States Agency for International Development (USAID) through FHI 360/Linkages [Grant Number: AID-OAA-A-14-00045]. Dr Sahay received the funds. PS and BG provided financial support for this study. "

6. Thank you for stating the following in the Funding Section of your manuscript: 

"This work was supported by the United States Agency for International Development (USAID) through FHI 360/Linkages [Grant Number: AID-OAA-A-14-00045]. The corresponding author received the funds. PS and BG provided technical and financial support for this study."

"This work was supported by the United States Agency for International Development (USAID) through FHI 360/Linkages [Grant Number: AID-OAA-A-14-00045]. Dr Sahay received the funds. PS and BG provided financial support for this study. "

7. In your Data Availability statement, you have not specified where the minimal data set underlying the results described in your manuscript can be found. PLOS defines a study's minimal data set as the underlying data used to reach the conclusions drawn in the manuscript and any additional data required to replicate the reported study findings in their entirety. All PLOS journals require that the minimal data set be made fully available. For more information about our data policy, please see http://journals.plos.org/plosone/s/data-availability.

Additional Editor Comments:

Kindly address the comments raised by the reviewers and submit the revised draft for final review.

Reviewers' comments:

Reviewer's Responses to Questions

**Comments to the Author**

1. Is the manuscript technically sound, and do the data support the conclusions?

Reviewer #1: Partly

Reviewer #2: Yes

Reviewer #3: Yes

2. Has the statistical analysis been performed appropriately and rigorously? 

Reviewer #1: N/A

Reviewer #2: Yes

Reviewer #3: N/A

3. Have the authors made all data underlying the findings in their manuscript fully available?

Reviewer #1: No

Reviewer #2: No

Reviewer #3: Yes

4. Is the manuscript presented in an intelligible fashion and written in standard English?

Reviewer #1: No

Reviewer #2: Yes

Reviewer #3: Yes

5. Review Comments to the Author

Reviewer #1: Congratulations on presenting an interesting project and manuscript. The English language used in the manuscript is difficult to read and follow by a native English speaker. You need this manuscript to be copy edited by a professional English writer so that it has further clarity and the important messages are not lost. Overall it is difficult to follow entirely due to some of the local terminology used and also the English expression which made reviewing the content challenging.

In relation to the actual manuscript:

Abstract - needs work:

Eight FGDs, 20 KIIs, and 20 IDIs - need to be spelled out on first appearance in the manuscript. These are copied above from the abstract but not defined.

The following should be moved from conclusions to results section in the abstract:

Primary barriers to health care access were lack of knowledge, myths and

misconceptions, not having faith in the quality of services, program invisibility in a rural

setting, and anticipated stigma at government health facilities.

Methods - this section could do with provision of a bit more detail in relation to the procedure undertaken during the recruitment of participants and the analysis of data. When was the analysis done? Was it carried out after all data collection was complete? Was there any interim analysis performed during data collection to ensure the emergent themes were explored fully or relevant further questions/prompts were included in subsequent data collection opportunities?

Participant recruitment also needs further description. Given this is an exceptionally difficult population to identify and then reach and subsequently recruit to the study, a detailed description of all the methods used would be extremely valuable for others intending on doing similar work within the population. What considerations did you make, how did you identify your participants? How can a reader be sure that coercion was not used? Please explain how you protected against this.

Results: Also needs re-writing to ensure the English language is grammatically correct and the expression is clear and concise. It was extremely difficult to follow all the descriptive findings and to understand the importance of each of the nuances to a 'non Indian' reader. Some of the data points were repeated which is unnecessary. I have copied one below as an example.

Discussion: The discussion was well thought through and much clearer than the rest of the manuscript. Highlighting particularly important issues - such as the extent of childhood sexual assault / abuse - is extremely important in the socio - cultural context and the structural systematic requirements prior to considering how HIV testing access can be improved. It is difficult for a foreigner to understand how sexual activity by a child aged 4 - 8 years can be described as 'debut' and even more difficult to understand how an inference of choice about sexual activity (and subsequent - safety of this) can be made when the child does not have capacity to refuse the sexual activity / assault in itself.

All other points raised in the discussion were appropriate and provided a strong argument for your conclusions.

Below are some specific content points that need addressing:

Ln 47 - disproportionately wedged by the HIV - what does wedged mean? This is not a familiar term.

Ln 51 - into a huge number - please provide an estimate - huge is a subjective term

Ln 226 - “About HIV I have only heard that it happens....now I do not know how it happens and why

227 it happens because I haven’t done anything like that for which I have to go to see a doctor,

228 if it was then I would have told you. no madam I haven’t heard, but yes I have read it in

229 the newspaper, otherwise, I haven’t heard of it and here the venereal disease doctor comes

230 from =Bhopal=, I have seen his advertisement in a newspaper, otherwise I haven’t heard

231 of it” [MP-R-ID-07, Madhya Pradesh] - these data have been used twice (also starts LN 283).

Reviewer #2: Thank you for the opportunity to review this interesting paper. The authors conducted a qualitative study in four states of India to identify barriers and enablers for men who have sex with men in using national HIV programmes. using grounded theory the authors conductor range of interviews to identify common themes related to cultural barriers low visibility and health system challenges and linking MSM two required HIV prevention programmes. The paper provides important findings button into India that advance the literature in HIV prevention I have identified some comments below which I encourage the authors to address in order to refine the presentation of the results

Reviewer #3: • An interesting and pertinent study on MSM in rural India and their challenges in accessing health care. Key populations face multiple social and access barriers that have been brought out effectively here.

• There is a typographical error in the abstract. P3 line 38: ‘a’ is not required in the line ‘(ASHAs), a frontline health workers for MSM.’

• While the title states ‘health care’, the article focuses on HIV related services. I would suggest that the title reflects that. Otherwise, the reader expects a broader discussion on health services and finds that missing.

• Introduction: The article begins with the HIV epidemic in India and the status of MSM as a key population. It would be good to add some more background on the nature of HIV in India, and the Government’s response. There is a mention of the ‘national program’ and of the TI and Link Workers, but which part of the picture they comprise is not clear. This could be challenging for a non-Indian reader to comprehend.

• Materials and methods: Some questions need to be addressed: On what basis were the districts selected? This needs to be added. Also, why were only rural samples considered? If the guides were developed in the local languages, how were they checked for uniformity in meaning across the three languages? What about the in-depth interview guide? Was it also developed in a similar manner? It would be added there. Are the ‘face to face interviews’ the same as in-depth interviews or key informant interviews? Would be good to broadly mention the themes covered by the interviews and FGDs. These were sensitive interviews and discussions, so then how was confidentiality and privacy maintained? A line on data management would be good to add, i.e., how was data protected (passwords, restricted access)?

• The results have been discussed adequately and well supported by quotes.

• Discussion: ‘…male rape and sexual abuse of child are not considered crimes’ – this sentence would require a reference. The acronyms HRGs and HRMSM need to be expanded at first use.

• Conclusion: There is much scope of strengthening the conclusion by listing out clearly the policy recommendations, as there are several important ones that emerge from the findings and have been mentioned in the discussion to some extent. Clearly there is a need for strengthening Government capacities and available systems for support, and this is a critical output of the analysis.

6. PLOS authors have the option to publish the peer review history of their article (what does this mean?). If published, this will include your full peer review and any attached files.

Reviewer #1: No

Reviewer #2: **Yes: **Dr Danish Ahmad

Reviewer #3: No

---

## [Author Response · Author response to Decision Letter 0]

20 Oct 2022

Point by point responses to reviewer’s comments

Thank you for your suggestion. We have ensured that our manuscript meets PLOS ONE's style requirements, including those for file naming. 

2. You indicated that you had ethical approval for your study. Please clarify whether minors were involved in your study. If so, in your Methods section, please ensure you have also stated whether you obtained consent from parents or guardians of the minors included in the study or whether the research ethics committee or IRB specifically waived the need for their consent.

Thank you for your point. There were no minors involved in the study. We have added the same to confirm on Page no. 09, line no.: 164 

3. Please describe in your methods section how capacity to provide consent was determined for the participants in this study. 

Thank you for your suggestion. As suggested we have described the informed consent process in the methods section. 

Page no: 12

Line no: 260-284

4. Please also state whether your ethics committee or IRB approved this consent procedure. 

Yes, our ethics committee/IRB had approved this consent procedure vide letter number NARI-EC/2014-13. We have mentioned this in methods. 

Page no: 12

Line no: 255-256

5. If you did not assess capacity to consent please briefly outline why this was not necessary in this case. 

Thank you for your suggestion. As suggested we have described the steps used while assessing the capacity of the participants to consent. 

Page no: 12

Line no: 260-284

6. Please provide the full names of all ethics committees that approved this study. 

Thank you. We have added the same 

Page no: 12

Line nos: 256-260

7. Thank you for stating the following financial disclosure: 

"This work was supported by the United States Agency for International Development (USAID) through FHI 360/Linkages [Grant Number: AID-OAA-A-14-00045]. Dr. Sahay received the funds. PS and BG provided financial support for this study. "

Thank you for your suggestions. We have deleted the funding information from the manuscript as instructed. We have mentioned this information in the cover letter. 

8. Thank you for stating the following in the Funding Section of your manuscript: 

"This work was supported by the United States Agency for International Development (USAID) through FHI 360/Linkages [Grant Number: AID-OAA-A-14-00045]. The corresponding author received the funds. PS and BG provided technical and financial support for this study."

"This work was supported by the United States Agency for International Development (USAID) through FHI 360/Linkages [Grant Number: AID-OAA-A-14-00045]. Dr. Sahay received the funds. PS and BG provided financial support for this study."

We have removed the funding information from the acknowledgment section of the manuscript accordingly. We have also removed the funding information from the funding section of the manuscript. 

Thank you for the opportunity. We have revised the funding statement Funding Statement section of the online submission form. The Funding Statement should be as follows- 

"This work was supported by the United States Agency for International Development (USAID) through FHI 360/Linkages [Grant Number: AID-OAA-A-14-00045]. Dr. Sahay received the funds."

We have included the amended the statement in our cover letter. Thank you for the above change in online submission form on our behalf.

9. In your Data Availability statement, you have not specified where the minimal data set underlying the results described in your manuscript can be found. PLOS defines a study's minimal data set as the underlying data used to reach the conclusions drawn in the manuscript and any additional data required to replicate the reported study findings in their entirety. All PLOS journals require that the minimal data set be made fully available. For more information about our data policy, please see http://journals.plos.org/plosone/s/data-availability.

We have added the information. The minimal data is fully available for anyone. We have added this information in manuscript. 

Page no: 35

Line no: 880-882

Reviewer 1 Comments 

10. Congratulations on presenting an interesting project and manuscript. The English language used in the manuscript is difficult to read and follow by a native English speaker. You need this manuscript to be copy edited by a professional English writer so that it has further clarity and the important messages are not lost. Overall it is difficult to follow entirely due to some of the local terminology used and also the English expression which made reviewing the content challenging. Thank you for your suggestion. We have tried to simplify the language and it has been copy edited by Retd. Professor in English, Head of the Department, Department of English, SK University, Dumka, Jharkhand, India.

We have added the same in the acknowledgement section.

Page no.: 5, 6, 7,10,12,13, 14, 15, 17, 21

Line no.: 63, 70-71, 79, 98, 107, 111, 116, 129-130, 207-208, 266-268, 297, 322, 344, 401, 405, 516

11. Abstract - needs work

We have revised the abstract. 

Page no.: 03

Line no.: 24-53

12. Eight FGDs, 20 KIIs, and 20 IDIs - need to be spelled out on first appearance in the manuscript. These are copied above from the abstract but not defined. Thank you for your suggestion. We have expanded the acronyms accordingly. 

Page no: 3

Line no: 28-29

13. The following should be moved from conclusions to results section in the abstract: Primary barriers to health care access were lack of knowledge, myths and misconceptions, not having faith in the quality of services, program invisibility in a rural setting, and anticipated stigma at government health facilities. 

As suggested we have moved the sentence in the results section. Thank you very much it has added strengths to the manuscript. 

Page no: 3

Line no:33-35

14. Methods - this section could do with provision of a bit more detail in relation to the procedure undertaken during the recruitment of participants. 

Thanks for suggestion. We have elaborated the recruitment procedure in the methods section. 

Page no: 9-10

Line nos: 171-191

15. Procedure for the analysis of data and When was the analysis done? Was it carried out after all data collection was complete? Was there any interim analysis performed during data collection to ensure the emergent themes were explored fully or relevant further questions/prompts were included in subsequent data collection opportunities? 

Thanks for pointing this out. Analysis was an ongoing process and coding was done by all the site investigators. Yes the interim analysis was performed to ensure the emerging themes. We have elaborated on the data analysis process in the manuscript. 

Page no: 11

Line nos: 237-239, 240-245

16. Participant recruitment also needs further description. Given this is an exceptionally difficult population to identify and then reach and subsequently recruit to the study, a detailed description of all the methods used would be extremely valuable for others intending on doing similar work within the population. 

We sincerely appreciate your comment that this was a conducted with a difficult and vulnerable population. We engaged the community and trained the research team into community literacy so that they really become cognizant to the issues of the vulnerability in the population being studied. We have elaborated the process of recruitment in the revised manuscript at - 

Page nos. 9-10

Line nos. 171-191

17. What considerations did you make, how did you identify your participants? 

Thank you for your suggestions. Participants were recruited with strict adherence to inclusion exclusion criteria of the protocol. 

18. How can a reader be sure that coercion was not used? Please explain how you protected against this. 

Thank you for your suggestion. We have added the details. 

Page no: 12-13

Line nos. 260-286

19. Results: Also needs re-writing to ensure the English language is grammatically correct and the expression is clear and concise. It was extremely difficult to follow all the descriptive findings and to understand the importance of each of the nuances to a 'non-Indian' reader. Some of the data points were repeated which is unnecessary. I have copied one below as an example.

As stated earlier the revised we have tried to simplify the language and it has been copy edited by Retd. Professor in English, Head of the Department, Department of English, SK University, Dumka, Jharkhand, India. We have added the same in the acknowledgement section.

Page no.: 5, 6, 7,10,12,13, 14, 15, 17, 21

Line no.: 63, 70-71, 79, 98, 107, 111, 116, 129-130, 207-208, 266-268, 297, 322, 344, 401, 405, 516

We have worked diligently on the quotations which were translated verbatim without changing the essence of the verbatim translations (data). We have used square brackets to explain the nuances to the 'non-Indian' readers. 

20. Discussion: The discussion was well thought through and much clearer than the rest of the manuscript. Highlighting particularly important issues - such as the extent of childhood sexual assault / abuse - is extremely important in the socio - cultural context and the structural systematic requirements prior to considering how HIV testing access can be improved. It is difficult for a foreigner to understand how sexual activity by a child aged 4 - 8 years can be described as 'debut' and even more difficult to understand how an inference of choice about sexual activity (and subsequent - safety of this) can be made when the child does not have capacity to refuse the sexual activity / assault in itself.

Thank you very much for your pointing out this important point. In Indian context also this is not sexual ‘Debut’ and therefore we have revised it to coercive sex/ past sexual abuse. However our data clearly shows that such young children slowly get into the practice of male to male sex. 

 Page no: 17, 28

Line no: 402-403, 701-702

21. Ln 47 - disproportionately wedged by the HIV - what does wedged mean? This is not a familiar term.

Thank you for your suggestion. We have used familiar term in the revised manuscript. 

Page No: 5

Line no: 62

22. Ln 51 - into a huge number - please provide an estimate - huge is a subjective term. 

We have added the estimated number of MSM as suggested.

Page No.: 5

Line no.: 66-67 

23. Ln 226 - “About HIV I have only heard that it happens....now I do not know how it happens and why

227 it happens because I haven’t done anything like that for which I have to go to see a doctor,

228 if it was then I would have told you. no madam I haven’t heard, but yes I have read it in

229 the newspaper, otherwise, I haven’t heard of it and here the venereal disease doctor comes

230 from =Bhopal=, I have seen his advertisement in a newspaper, otherwise I haven’t heard

Thank you for your suggestion. We have revised it to make it comprehensive.

Page no: 19

Line no: 441-447

24. 231 of it” [MP-R-ID-07, Madhya Pradesh] - these data have been used twice (also starts LN 283). 

We have deleted the duplicate quote.

Reviewer 2 

Thank you for the opportunity to review this interesting paper. The authors conducted a qualitative study in four states of India to identify barriers and enablers for men who have sex with men in using national HIV programmes using grounded theory the authors’ conducted range of interviews to identify common themes related to cultural barriers low visibility and health system challenges and linking MSM two required HIV prevention programmes. The paper provides important findings button into India that advance the literature in HIV prevention I have identified some comments below which I encourage the authors to address in order to refine the presentation of the results. 

25. Line 51, page 11: The underlying assumption that MSM are driving HIV incidence in India is challenging considering the high rates of blood borne routes that predominate in the North East.

Thank you for your comment. We have referred to the concentrated epidemic in India and justified our assumption for clarity in the revised manuscript. 

Page no: 5

Line nos.: 69-72

26. In this context, can the authors provide additional details of the current HIV status In India Moreover the reference quoted ref(13) is a 2012 NACO report which does not indicate current status.

We have updated the current HIV status in India in the revised manuscript as suggested. 

Page no: 5

Line nos.: 74-75

We have added the reference. 

Page no: 5

Line no: 75, Ref. no.: 6

27. The framing/rationale of the study aim, and context requires strengthening by highlighting important epidemiological and public health information from India in the background. 

As suggested, we have highlighted epidemiological and public health situation in the background section. 

Page no: 5

Line no: 66-67, 72-75

28. HIV prevalence among MSM needs to be situated first across overall HIV prevalence and incidence in India with reference to India’s progress over the MDG to meet the global HIV 90-90-90 targets.

We have moved overall HIV prevalence and incidence in India with reference to India’s progress over the MDG to meet the global HIV 95-95-95 targets

Page no: 6

Line no: 100-102

29. Lines 72: page 12, Please provide an overview of India’s HIV prevention national program in brief for international readers. 

As suggested, we have added the information on the National AIDS Control Organization in the revised manuscript.

Page no: 5

Line no: 77-97

30. Please explain what the link worker scheme is referenced in line 72.

LWS in rural areas focuses on demand generation for various HIV/AIDS-related services, linking target populations to existing services, and providing a stigma-free environment for sustained access to information and services under the National AIDS Prevention and Control Programme of India. We have explained it in the background.

Page no: 6

Line no: 94-97

31. In lines 85, please say what the HSS HIV prevalence rates for the study sites are providing key context is important here. 

As suggested, we have added HSS HIV prevalence rates for the study sites. It has added strength to the manuscript. Thank you very much. 

Page no: 8

Line no: 150-154

32. Lines 91, 13: While retaining confidentiality, please provide some description of who the community liaison and key population health leaders were. 

Community liaison represented the study community. Key population health leaders belonged to Community Based Organizations, Program personnel and clinicians providing health care to the study community. We have revised the manuscript accordingly at.

Page no: 9

Line no: 182-183

33. Currently, the paper does not distinguish between the study areas despite known development, cultural and ethnical variables acting in each state which may differently impact MSM’s engagement with the program. As part of the methods, the authors mention study areas however, can they please provide a break up of exact data collection numbers split across the study areas and across key respondents. 

We agree with the reviewers and hence as suggested, we have split across the study areas and across key respondents in the revised manuscript. 

Page no: 9-10

Line no: 170-171, 192-197

34. Understanding the results requires reference to the specific study sites and areas from where data was collected, are the authors able to identify common themes relevant to individual states from where data was collected in order to provide meaningful recommendation to policy makers.

This is a very valuable suggestion and it gives us confidence because we are already working on our next manuscript focusing on site-wise differences. It will definitely focus on cultural diversity and similarities. Very initial point we have added in the manuscript but we would like to bring it as a new manuscript. Since this is not conclusive we are not mentioning it as policy recommendation. 

Page no: 26

Line no: 646-650

35. Where possible use newer sources of data/literature. Ref 13 for example is the 2012 NACO report which has since been update. Please review the references for formatting issues. Currently, the author names and key headings for many references have errors for example reference 12 (Trust IHA) is incorrect 16 has text formatting issues’ 17 UNAIDS needs to be in full form, Ref 20 - text formatting errors.These are only some that I am sharing here for illustration, please review the full reference list for consistency.

We apologize. We have updated the references where possible and made appropriate formatting changes. 

Reviewer 3

An interesting and pertinent study on MSM in rural India and their challenges in accessing health care. Key populations face multiple social and access barriers that have been brought out effectively here.

Abstract There is a typographical error in the abstract.

36. P3 line 38: ‘a’ is not required in the line ‘(ASHAs), a frontline health workers for MSM.’

We apologize for the typographical error. We have corrected the error. 

Page no 4

Line no 46

37. While the title states ‘health care’, the article focuses on HIV related services. I would suggest that the title reflects that. Otherwise, the reader expects a broader discussion on health services and finds that missing.

Thank you for pointing it out. We have modified the title. 

Page no 1

Line no 1

38. Introduction The article begins with the HIV epidemic in India and the status of MSM as a key population. It would be good to add some more background on the nature of HIV in India, and the Government’s response. There is a mention of the ‘national program’ and of the TI and Link Workers, but which part of the picture they comprise is not clear. This could be challenging for a non-Indian reader to comprehend.

Thank you for your suggestion. We have added the background on the nature of HIV in India, and the Government’s response. Additionally we have expanded on the Link Workers Scheme in the revised manuscript for better comprehension of a non-Indian reader. 

Page No: 5, 6

Line no: 65-75, 777-98

39. Materials and methods: Some questions need to be addressed: 

On what basis were the districts selected? This needs to be added. 

In each region, the states were selected based on the prevalence of HIV/AIDS among the MSM population reported in HIV Sentinel Surveillance (HSS) and in consultation with the national program. We have added the information in the revised manuscript. 

Page No: 8

Line no:145-154

Also, why were only rural samples considered? 

Thank you for pointing it out. The study was conducted in urban and rural settings in selected states. However the data for health seeking behavior was presented for only rural setting. 

Page no: 8

Line no: 140-144

If the guides were developed in the local languages, how were they checked for uniformity in meaning across the three languages? 

The next manuscript focuses on comparison of urban and rural settings. We apologize, we developed the guides in English language first which were translated in local vernacular languages. The Principal Investigators of the sites compared the both English and vernacular version for uniformity and meanings. We have explained this in the methods section

Page No: 10

Line no: 199-206

What about the in-depth interview guide? Was it also developed in a similar manner? It would be added there. 

Yes, in-depth interviews were developed in similar manner. 

Page No: 10

Line no: 199-206

Are the ‘face to face interviews’ the same as in-depth interviews or key informant interviews? 

Yes. We conducted all interviews face to face. There were no telephonic interviews. 

Page No: 11

Line no: 229-230

Would be good to broadly mention the themes covered by the interviews and FGDs. 

Using grounded theory approach we have mentioned the broad interviews topic in KII, IDI and FGDs. 

Page No: 10

Line no: 206-208

These were sensitive interviews and discussions, so then how was confidentiality and privacy maintained? 

Thank you for your suggestion. We have added the details in the methods section

Page No: 13

Line no: 287-292

A line on data management would be good to add, i.e., how was data protected (passwords, restricted access)?

Thank you for your suggestion. We have added the details in the methods section

Page No: 13

Line no: 291-292

40. The results have been discussed adequately and well supported by quotes.

Thank you very much. We appreciate the encouragement. 

41. Discussion ‘…male rape and sexual abuse of child are not considered crimes’ – this sentence would require a reference. The acronyms HRGs and HRMSM need to be expanded at first use. 

We have the reference as suggested at Page No: 27-28; Line no: 671-683

42. Conclusion There is much scope of strengthening the conclusion by listing out clearly the policy recommendations, as there are several important ones that emerge from the findings and have been mentioned in the discussion to some extent. Clearly there is a need for strengthening Government capacities and available systems for support, and this is a critical output of the analysis. 

As suggested, we have added policy recommendations in the conclusion. 

Page No: 33

Line no: 829-844

---

## [Decision Letter · Decision Letter 1]

26 Dec 2022

PONE-D-22-17552R1Exploring access to HIV-related services and programmatic gaps for Men having Sex with Men (MSM) in rural India- a qualitative study.PLOS ONE

Dear Dr. Sahay,

Thank you for submitting your manuscript to PLOS ONE. After careful consideration, we feel that it has merit but does not fully meet PLOS ONE’s publication criteria as it currently stands. Therefore, we invite you to submit a revised version of the manuscript that addresses the points raised during the review process.

We look forward to receiving your revised manuscript.

Kind regards,

Jalandhar Pradhan

Academic Editor

PLOS ONE

Journal Requirements:

Additional Editor Comments:

One of the Reviewers still feel some of the comments are not yet addressed. So, I request the authors to revise the draft accordingly.

Reviewers' comments:

Reviewer's Responses to Questions

**Comments to the Author**

1. If the authors have adequately addressed your comments raised in a previous round of review and you feel that this manuscript is now acceptable for publication, you may indicate that here to bypass the “Comments to the Author” section, enter your conflict of interest statement in the “Confidential to Editor” section, and submit your "Accept" recommendation.

Reviewer #1: All comments have been addressed

Reviewer #2: (No Response)

Reviewer #3: All comments have been addressed

2. Is the manuscript technically sound, and do the data support the conclusions?

Reviewer #1: Yes

Reviewer #2: No

Reviewer #3: Yes

3. Has the statistical analysis been performed appropriately and rigorously? 

Reviewer #1: N/A

Reviewer #2: Yes

Reviewer #3: N/A

4. Have the authors made all data underlying the findings in their manuscript fully available?

Reviewer #1: Yes

Reviewer #2: Yes

Reviewer #3: Yes

5. Is the manuscript presented in an intelligible fashion and written in standard English?

Reviewer #1: Yes

Reviewer #2: Yes

Reviewer #3: Yes

6. Review Comments to the Author

Reviewer #1: Thank you for your attention to detail and the issues raised in my previous review of the original manuscript. You have satisfactorily addressed my previous comments. This paper addresses many important aspects of rural Indian HIV service gaps and societal underpinnings that require immediate attention. Congratulations on bringing such an important piece to international attention.

Reviewer #2: A much-improved version is submitted, which clarifies prior comments. Again this is a paper of great public health importance, and I would be pleased to see it published soon. The current revision still requires language editing for clarity and adherence to an academic style guide. For example, please spell out numbers below 10 instead of using 4,5 etc., as done in places. The second key comment is for authors to review the literature-results-discussion sections to provide better linkages in current programmatic gaps in the study sites that led to the study. I understand in the literature section prevalence of MSM with HIV. MSM is a factor as is anonymity and sociocultural issues in rural areas, but can the authors please review the literature section and provide clearer links(rationale) for the study related to rural areas given that NACO certainly has a rural component in place where health services are active. For example, you can contrast NACO"s success in rural areas in other states and perhaps state that the three study sites are also considered part of socio-economically disadvantaged states. As there is a large co-authorship for the paper, I would like the paper to be reviewed for clarity in the communication of the key messages across the three sections that I have identified above. The conclusion also needs to be reworded as currently, it reads like a programmatic implications section which fits better under discussion. The authors may like to read the literature and go straight to discussion to see how the paper flows as a tip

Reviewer #3: The revised manuscript addresses all my previous comments satisfactorily. In some places there are minor language errors, and therefore I recommend a thorough re-checking of the text, otherwise in terms of content I do not have any further suggestions.

7. PLOS authors have the option to publish the peer review history of their article (what does this mean?). If published, this will include your full peer review and any attached files.

Reviewer #1: No

Reviewer #2: **Yes: **DR Danish Ahmad MBBS,MSc,PhD

Reviewer #3: No

---

## [Author Response · Author response to Decision Letter 1]

9 Feb 2023

Point by point responses to reviewer’s comments

Reviewer #1: 

Comment: Thank you for your attention to detail and the issues raised in my previous review of the original manuscript. You have satisfactorily addressed my previous comments. This paper addresses many important aspects of rural Indian HIV service gaps and societal underpinnings that require immediate attention. Congratulations on bringing such an important piece to international attention. 

Response: Thank you. We are grateful.

Reviewer #2: 

Comment: A much-improved version is submitted, which clarifies prior comments. Again this is a paper of great public health importance, and I would be pleased to see it published soon.

The current revision still requires language editing for clarity and adherence to an academic style guide. For example, please spell out numbers below 10 instead of using 4, 5 etc., as done in places. 

Response: Thank you for your guidance. We have corrected the language editing for clarity adhering to APA academic style. 

Page nos: 3, 8, 9, 11.16.17.19, 26, 28

Line nos: 29, 30,163, 166, 178, 188, 189, 238, 349, 357, 366, 378, 379,383, 427, 580, 625 (Line numbers as per track changed copy with all mark-up setting)

Comment: The second key comment is for authors to review the literature-results-discussion sections to provide better linkages in current programmatic gaps in the study sites that led to the study. I understand in the literature section prevalence of MSM with HIV. MSM is a factor as is anonymity and sociocultural issues in rural areas, but can the authors please review the literature section and provide clearer links (rationale) for the study related to rural areas given that NACO certainly has a rural component in place where health services are active. For example, you can contrast NACO’s success in rural areas in other states and perhaps state that the three study sites are also considered part of socio-economically disadvantaged states.

Response: We agree with the reviewer regarding NACO’s rural component being active. We have accordingly included the literature in the introduction section. 

Page nos: 5, 6

Line nos: 83-86, 97-102 (Line numbers as per track changed copy with all mark-up setting)

We are thankful for this suggestion as it has provided strong rationale for including particular study sites. 

Comment: As there is a large co-authorship for the paper, I would like the paper to be reviewed for clarity in the communication of the key messages across the three sections that I have identified above. 

Response: As suggested we have circulated the revised version to all authors for their review and suggestions if any. 

Comment: The conclusion also needs to be reworded as currently, it reads like a programmatic implications section which fits better under discussion. The authors may like to read the literature and go straight to discussion to see how the paper flows as a tip 

Response: Thank you for your suggestions. We have shifted the conclusion points to discussion section. We have accordingly revised the conclusion. 

Page no: 32-33

Line no: 732-746 

Reviewer #3: 

Comment: The revised manuscript addresses all my previous comments satisfactorily. In some places there are minor language errors, and therefore I recommend a thorough re-checking of the text, otherwise in terms of content I do not have any further suggestions. 

Response: Thank you for reviewing the manuscript. We have carefully gone through the manuscript for English language editing and tried our best to remove any error.

---

## [Decision Letter · Decision Letter 2]

12 Apr 2023

Exploring access to HIV-related services and programmatic gaps for Men having Sex with Men (MSM) in rural India- a qualitative study

PONE-D-22-17552R2

Dear Dr. Sahay,

We’re pleased to inform you that your manuscript has been judged scientifically suitable for publication and will be formally accepted for publication once it meets all outstanding technical requirements.

Kind regards,

Nelsensius Klau Fauk, S.Fil., M., MHID, MSc, PhD

Academic Editor

PLOS ONE

Additional Editor Comments (optional):

Reviewers' comments:

Reviewer's Responses to Questions

**Comments to the Author**

1. If the authors have adequately addressed your comments raised in a previous round of review and you feel that this manuscript is now acceptable for publication, you may indicate that here to bypass the “Comments to the Author” section, enter your conflict of interest statement in the “Confidential to Editor” section, and submit your "Accept" recommendation.

Reviewer #2: All comments have been addressed

2. Is the manuscript technically sound, and do the data support the conclusions?

Reviewer #2: Yes

3. Has the statistical analysis been performed appropriately and rigorously? 

Reviewer #2: N/A

4. Have the authors made all data underlying the findings in their manuscript fully available?

Reviewer #2: (No Response)

5. Is the manuscript presented in an intelligible fashion and written in standard English?

Reviewer #2: Yes

6. Review Comments to the Author

Reviewer #2: Thank you for revising the paper and addressing my comments, I am recommnding publication and would ask the journal to support an expediated process to publication to make these important findings available soon

7. PLOS authors have the option to publish the peer review history of their article (what does this mean?). If published, this will include your full peer review and any attached files.

Reviewer #2: **Yes: **DR Danish Ahmad MBBS(Delhi Uni.),MSc(Oxford Uni.).PhD(Canberra Uni) MNAMS(India),UKFPH

---

## [Editor Report · Acceptance letter]

24 Apr 2023

PONE-D-22-17552R2 

Exploring access to HIV-related services and programmatic gaps for Men having Sex with Men (MSM) in rural India- a qualitative study 

Dear Dr. Sahay:

I'm pleased to inform you that your manuscript has been deemed suitable for publication in PLOS ONE. Congratulations! Your manuscript is now with our production department. 

Kind regards, 

on behalf of

Dr. Nelsensius Klau Fauk 

Academic Editor

PLOS ONE